# Work Disabling Nerve Injury at Both Elbows Due to Laptop Use at Flexible Workplaces inside an Office: Case-Report of a Bilateral Ulnar Neuropathy

**DOI:** 10.3390/ijerph17249529

**Published:** 2020-12-19

**Authors:** P. Paul F. M. Kuijer, Joris van der Pas, Henk F. van der Molen

**Affiliations:** 1Department of Public and Occupational Health, Netherlands Centre for Occupational Diseases, Amsterdam Public Health Research Institute, Amsterdam UMC, University of Amsterdam, 1105 Amsterdam, The Netherlands; h.f.vandermolen@amsterdamumc.nl; 2Self-Employed Occupational Physician, 5221 Den Bosch, The Netherlands; info@vanderpasbedrijfsarts.nl

**Keywords:** occupational disease, neuropathy, laptop, risk factors, workstation, prevention, elbow, nerve injury, ergonomics

## Abstract

Background: This case report describes whether a female civil servant who developed bilateral ulnar neuropathy can be classified as having an occupational disease. Methods: The Dutch six-step protocol for the assessment and prevention of occupational diseases is used. Results: Based on the six-step protocol, we propose that pressure on the ulnar nerve in the elbow region precipitated the neuropathy for this employee while working prolonged periods in elbow flexion with a laptop. Conclusion: Despite the low incidence laptop use might be a risk factor for the occurrence of ulnar neuropathy due to prolonged pressure on the elbow. Employers and workers need to be educated about this disabling occupational injury due to laptop use and about protective work practices such as support for the upper arm and elbow. This seems especially relevant given the trend of more flexible workspaces inside and outside offices, and given the seemingly safe appearance of laptop use.

## 1. Introduction

Health and safety professionals might be confronted with occupational injuries and diseases in their work, forcing them to seek associated risk factors and take preventive actions. Knowledge of the occurrence of these injuries and diseases, explanatory models, and measurements at worksites can be helpful in order to recognize the safety hazards and health risk. Occupational injuries are characterized by a body lesion at the organic level resulting from acute exposure to energy (mechanical, thermal, electrical, chemical or radiant) in a work environment in amounts that exceed the threshold of physiological tolerance [1]. In contrast, occupational diseases are often characterized by cumulative exposures and longer latency before the disease emerges [2]. An example of an occupational disease caused by prolonged mechanical exposure in the workplace is ulnar neuropathy at the elbow, also referred to as Cubital Tunnel Syndrome. Ulnar neuropathy at the elbow is the second most common form of nerve entrapment after Carpal Tunnel Syndrome (CTS) of the wrist [3].

For clinically diagnosed ulnar neuropathy at the elbow, a prevalence of 0.6–0.8% has been found in a French working population [4], and 1.8–5.9% based on self-reports using validated questionnaires among a cohort of adult residents of the St. Louis metropolitan area [5]. Incident rates of 20–30 per 100,000 person-years have been reported for general population samples in Italy, the United Kingdom, and the United States of America [6,7,8]. Among patients with ulnar neuropathy, 61% still had symptoms and 45% still had disability after a median follow-up time of 4 years [9]. This disease is associated with high-force jobs [10,11,12,13]. However, the present case study describes a Dutch female civil servant developing a bilateral ulnar neuropathy after prolonged pressure on the flexed elbows while working with a laptop at an office with flexible workplaces.

In the Netherlands, occupational physicians are obliged by law to notify cases of occupational disease to the National Register of the Netherlands Center for Occupational Diseases. Each case of an occupational disease is anonymously reported to the Netherlands Center for Occupational Diseases, with the following information recorded in its database: disease or pathology with clinical diagnosis, demographic characteristics, work-related exposure, job title, economic sector, and incapacity for work. In contrast to most other countries, there is no financial compensation system for diagnosed occupational diseases in the Netherlands. The Netherlands Center for Occupational Diseases does not have a list of occupational diseases. Any disease can be recognized as an occupational disease if the causality is sufficiently proven, preferably using the six-step protocol [1,14,15]. In line with the Dutch law, a disease is recognized as an occupational disease if the work-related fraction is estimated to be more than 50%. Therefore, the aim of this study is to systematically describe whether this case of ulnar neuropathy can be classified as an occupational disease due to laptop use.

## 2. Materials and Methods 

We will describe this case on the basis of the six-step protocol for the assessment and prevention of occupational diseases, as is used in the Netherlands [1,14,15], and also add the CARE checklist for describing case reports in the Appendix A. The six steps are as follows:Step 1. Diagnosis of the disease;Step 2. Evidence from the literature regarding work-relatedness;Step 3. Assessment of the nature and level of work-related exposure;Step 4. Non-work-related risk factors and individual susceptibility;Step 5. Assessment of occupational disease;Step 6. Treatment and preventive measures.

The patient involved provided inform consent after reading the final version of the manuscript. 

## 3. Results 

### 3.1. Step 1: Diagnosis

A 59-year-old female civil servant contacted her company doctor with a 2-month history of distal sensory and motor symptoms that significantly affected her activities of daily living and work ability. She experienced typical paresthesias involving digits IV and V of both hands, as well as a loss of grip strength in both hands.

On presentation, atrophy of m. adductor pollicis and intrinsic hand muscles on both hands was found. There was an evident loss of grip strength in both hands. Additionally, the spreading and closing of the fingers was subjectively impaired. Froment’s sign was positive (right > left hand). Tinel’s sign was negative on both sides.

She was referred to a neurologist, who made the diagnosis of bilateral ulnar neuropathy on the basis of symptoms and signs. The clinical diagnosis was confirmed by electroneurography (ENG), which showed a severe conduction block across both elbows.

The patient was referred to a hand therapist, who measured a grip strength of 6 kg on the left hand, and 5 kg on the right hand, using a Jamar Hand Dynamometer in position 2. Additionally, the pinch strength in the lateral grip was found to be below average on both sides (1.5 kg).

### 3.2. Step 2: Evidence from the Literature Regarding Work-Relatedness

To assess the evidence regarding a relationship between computer work and ulnar neuropathy, two searches were performed in Pubmed on 3 December 2020 using the search terms: (1) “compression, ulnar nerve [MeSH Terms] AND diseases, occupational [MeSH Terms]” and (2) “compression, ulnar nerve [MeSH Terms] AND work”. The first search resulted in 27 studies and the second in 57 studies. Furthermore, Web of Science and UpTodate was used. We included thirteen original studies that addressed the topic of work and ulnar neuropathy [12,13,16,17,18,19,20,21,22,23,24,25,26]. In addition, a review was included [27]. Three studies described the relationship between computer work and ulnar neuropathy: a double case-referent study [16], a cross-sectional study [19] and a multicenter case-control study [26]. The case-referent study compared patients with clinically assessed ulnar neuropathy by ENG (n = 546) and patients with ulnar neuropathy-like symptoms (n = 633) with three age- and sex-matched community referents per case [16]. This study found a negative exposure–response relationship between hours of daily computer use and ulnar neuropathy, and the same tendency was seen for ulnar neuropathy-like symptoms. In contrast, leaning on the elbow while working showed a significantly higher risk for clinically assessed ulnar neuropathy (OR 2.16, 95% CI 1.06–4.44), but not for ulnar neuropathy-like symptoms. The cross-sectional study described four radiologists diagnosed with cubital tunnel syndrome (two (50%) unilateral, two (50%) bilateral) [19]. The four spent 3.4 ± 0.3 years (mean ± standard error of the mean) as staff radiologists in their filmless department, performing computer keyboard and mouse or trackball image manipulation and work list navigation, typing preliminary reports and telephone notifications, and editing electronically and approving dictated final reports. All four are academically active and had significantly greater workday hours (*p* < 0.05) and performed more research (*p* < 0.003) than the eight asymptomatic radiologists. Sonography was ruled out as a risk factor in these cases. The multicenter case-control study consecutively enrolled 220 cases and 460 controls among participants admitted to four electromyography labs. The diagnosis was made on clinical and neurographic findings. The control group included all other participants without signs/symptoms of ulnar neuropathy and with normal ulnar nerve neurography. The multivariable analysis showed that ulnar neuropathy at the elbow was associated with male sex (OR = 2.4, 95%CI = 1.6–3.7), smoking habits (>25 pack-years (OR = 2.3, 95%CI = 1.3–4.1)), body mass index (OR = 1.05, 95%CI 1.01–1.10), polyneuropathies (OR = 4.1, 95%CI 1.5–11.5), and leaning with flexed elbow on a table/desk (OR = 1.5, 95%CI 1.0–2.2). The percentages of cases (57%) and controls (53%) did not differ as regards their self-reporting of ‘laying the pronated forearm on the edge of a hard surface (e.g., when using computer or sitting at the table/desk)’. So, in summary, leaning on the elbow can be seen as a significant risk factor for the work-relatedness of ulnar neuropathy.

### 3.3. Step 3: Nature and Level of Work-Related Exposure

For the previous four years, the civil servant had worked at her own computer workstation with a desktop computer. Due to a shortage of office space, her employer introduced flexible workspaces. In this context she was facilitated with a laptop, without a separate keyboard or mouse (Figure 1). The job required computer use for most of the time, and therefore more than two hours a day on average. The computer time did not change while working with the laptop. She worked four days a week at different workspaces.

Within the course of several months prior to presentation, she reported working in such a manner that prolonged pressure was applied on both elbows. After five months she started to experience paresthesias, involving digits IV and V of both hands, as well as a loss of grip strength in both hands.

### 3.4. Step 4: Non-Work-Related Risk Factors and Individual Susceptibility

There was no past medical history of other possible risk factors, such as obesity, trauma, intoxications due to, e.g., drug, alcohol or tobacco consumption, medication use, neurological disorders, diabetes mellitus or other upper-limb work-related musculoskeletal disorders. She reported no strenuous activities (hobbies, sports) outside of work, or laptop use at home, or injuries which could have contributed to the onset or worsening of symptoms.

### 3.5. Step 5: Assessment of Occupational Disease

In the Netherlands, a disease is classified as an occupational disease if the work-related fraction is estimated to be more than 50% [1,14]. Moreover, in the Netherlands no financial compensation is obtained for reporting an occupational disease to the Netherlands Center for Occupational Diseases. In the assessment of an occupational disease, most clinicians focus primarily on three of the Hill criteria, namely a high relative risk, temporality and the presence of a biological gradient [2]. As described in step 2, an OR higher than 2 was found for leaning on the elbow for clinically assessed ulnar neuropathy. As a rule of thumb, this implies an attributable risk of more than 50%. Therefore, a high relative risk seems present. More importantly, no other non-work-related or personal risk factors with a similar risk were present. Temporality is supported by the onset of complaints related to the change from working behind her own computer workstation to working at her laptop without a separate keyboard and mouse at different workstations. The biological gradient is the biomechanical pressure on the ulnar nerve while working with the flexed elbow at the table [13,26,27,28,29,30]. We feel that specific activities, for instance, repetitive copying and pasting using the CTRL-V and CTRL-C keys, might increase the risk of ulnar nerve compression. The time for the onset of complaints (within several months) also seems in line with this criterion. Therefore, the ulnar neuropathy was classified as an occupational disease and reported to the Netherlands Center for Occupational Diseases.

### 3.6. Step 6: Treatment and Prevention

Based on his findings, the neurologist advised nerve surgery (neurolysis). The patient declined this type of treatment and opted for conservative management. The treatment consisted of training muscles, posture advice and nerve gliding exercises. In due course, a slow improvement in self-reported symptoms and strength was observed.

A workplace investigation was conducted by an ergonomist to determine the working conditions and to advance return to work. On the basis of this investigation, recommendations were made for accommodation, including the use of RSI software, speech recognition software, a headset, and a separate mouse, keyboard and display. For prevention, forearm supports might also be effective [31]. After six months a gradual return to work was realized. In the absence of negative prognostic factors (excluding female sex), such as smoking, obesity, distal upper-extremity fractures, a relatively favorable prognosis is to be expected [9]. The reduction in exposure and the subsequent improvement of symptoms also supports the temporality, as is described in step 5.

## 4. Discussion

In the Netherlands, there is no financial compensation system for diagnosed occupational diseases and no list of occupational diseases. Financial compensation in the first year of sick leave is independent of the work-relatedness of a disease. Any disease can be recognized as an occupational disease if the causality is sufficiently proven, preferably using the six-step protocol [1,14,15]. Therefore, the case definitions, exposure criteria and the assessment procedure might be different in other countries given the main purpose of prevention. By Dutch law, the occupational physician is the only medical specialist that is qualified to report a disease as an occupational disease. Therefore, the case presented in our paper might not be qualified as an occupational disease in other countries or according to other medical specialists.

Nowadays, traditional office-based computer work is shifting more and more towards laptop work anywhere imaginable, with ever-present access to data over the internet. “Anywhere imaginable” is most often in a conference room or at fixed-height tables [32]. Thereby, the laptop is primarily used for documents, data, email, calendar and internet browsing on a daily basis. These workers are working under increasingly less favorable ergonomic conditions compared to traditional office-based computer work, mostly with adaptations of the workstation. As expected, a recent review showed that screen work resulted in an increased risk of musculoskeletal symptoms, such as self-reported neck, shoulder and distal upper extremity symptoms and diagnosed carpal tunnel syndrome. However, the evidence is heterogeneous, and remarkably the review found no studies related to laptop use [33]. In addition, laptop use is associated with other specific disorders, such as thermal burns on the lower limbs [34,35], decreased human sperm quality [36], and partial amputation of the foot [37]. 

In summary, despite the low incidence of ulnar neuropathy, there might be an increasing proportion of workers at risk for this work-disabling occupational disease due to laptop use at flexible workplaces inside and outside offices. Protective measures should include, among other things, support of the forearms and elbows [31].

## 5. Conclusions

Laptop use might be a risk factor for the occurrence of ulnar neuropathy due to prolonged pressure on the elbow, although this is not previously described in the medical literature. Protective measures should include support of the forearms and elbow, as well as the reduction in the long-term and predominant unilateral use of a mouse or keyboard, and the more frequent alternation of tasks. Employers and workers need to be informed about the specific health risks of laptop use and about protective work practices, particularly in view of the increasing number of people working unbound by time and place.

## Figures and Tables

**Figure 1 ijerph-17-09529-f001:**
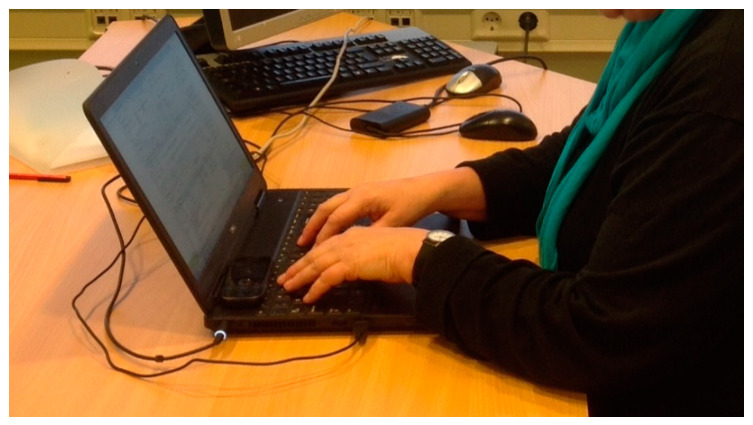
The 59-year-old female civil servant showing how she worked with her laptop when symptoms significantly affected her activities of daily living and work after she had returned to work.

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
