# Peer review of "Work Disabling Nerve Injury at Both Elbows Due to Laptop Use at Flexible Workplaces inside an Office: Case-Report of a Bilateral Ulnar Neuropathy"

_ijerph, 2020, doi:10.3390/ijerph17249529_

Round 1

Reviewer 1 Report

Interesting case because it is a frequent question in our country (not Netherland). However, I feel some components are missing

  1. I feel this is a case report not a review. A good case report integrates a review .
  2. To be considered as a case report, authors should probably follow the CARE guidelines (checklist appropriate for case report)
  3. I like the idea of the 6 steps. However, here the question is international readership, and follow the standard of their own country, reasons should be given in detail in the introduction and discussion
  4. I have a very important concern about the description as a clinician: authors described a very severe with muscular atrophy in the ulnar nerve distribution, with 2-month history of symptoms only? I think the step 4 is only possible after:
    1. What are the other factors, including weight/ BMI (obesity is one cause of it independent of work), endocrine status, glycemia, familial factor of pressure hypersensitivity, toxic neuropathy (drug, tobacco, alcohol consumption)
    2. Differential diagnosis is very important to be given and ruled out explicitly, with complete detail of the exam of upper extremity, association with other nerve entrapment (including thoracic outlet syndrome and carpal tunnel syndrome, “ double-crush syndrome”), pressure hypersensitivity.
    3. Nerve conduction results with curves should be also given
    4. The possible anatomic variations have been probably considered by the surgeon, and the MRI is necessary to be given also
  5. When look at the figure 1, we could see a regular keyboard, why using the laptop? Ar you sure there is a pressure in the ulnar tunnel in such patient because when you look the figure, she seems to press the posterior medial portion of the elbow, not the ulnar tunnel (more anterior).

Publishing case report is  difficult because differential should be ruled extensively, as well as getting compensation for patient.

Minor typo=> L91[ ….. -25] not         [ …..-25)

Author Response

Reviewer 1 report:
Comments and Suggestions for Authors
Interesting case because it is a frequent question in our country (not Netherland). However, I feel some components are missing

  •  First of all, we like to thank the reviewer for the time taken to critically review our paper and the compliment regarding the relevance. The useful comments have improved the quality of our paper.

1. I feel this is a case report not a review. A good case report integrates a review.

  •  We agree with the reviewer and have therefore deleted review from the subtitle, page 1, line 1 and 5.
    Moreover we have updated our integrated review to December 3rd 2020, page 2, line 88. One additional paper was included [26] also providing evidence for ‘leaning with flexed elbow on a table/desk’, page 3, lines 92-95 and 108-119.

2. To be considered as a case report, authors should probably follow the CARE guidelines (checklist
appropriate for case report)

  •  Good suggestion and we mentioned the use of the checklist in our methods section and have added the checklist as appendix, page 2, line 61-62

3. I like the idea of the 6 steps. However, here the question is international readership, and follow the
standard of their own country, reasons should be given in detail in the introduction and discussion

  •  We have added the rationale both in the introduction page 2, line 55 and added an extra paragraph in the discussion regarding the standard in the Netherlands compared to other countries to page 5, line 172-180.

4. I have a very important concern about the description as a clinician: authors described a very severe with muscular atrophy in the ulnar nerve distribution, with 2-month history of symptoms only? I think the step 4 is only possible after:

1. What are the other factors, including weight/ BMI (obesity is one cause of it independent of work), endocrine status, glycemia, familial factor of pressure hypersensitivity, toxic neuropathy (drug, tobacco, alcohol consumption)
2. Differential diagnosis is very important to be given and ruled out explicitly, with complete detail of the exam of upper extremity, association with other nerve entrapment (including thoracic outlet syndrome and carpal tunnel syndrome, “ double-crush syndrome”), pressure hypersensitivity.
3. Nerve conduction results with curves should be also given
4. The possible anatomic variations have been probably considered by the surgeon, and the MRI is necessary to be given also

  •  Thank you for your suggestions. We can rule out obesity and toxic neuropathy due to drug, tobacco, or alcohol consumption and added these to the other mentioned non-work-related causes like trauma, medication use, neurological disorders, diabetes mellitus or other upper-limb work-related musculoskeletal disorder, page 5 line 135-136. Unfortunately we are not able to provide the other
    requested data. This is partly due the level of detail required in occupational disease reporting as is described in the revised discussion (page 5, line 172-180) and that the patient now receives her occupational health care form another occupational health service and given the Dutch privacy regulations we feel that it is not appropriate to contact the patient not more often than already is
    done for this case report.

5. When look at the figure 1, we could see a regular keyboard, why using the laptop? Ar you sure there is a pressure in the ulnar tunnel in such patient because when you look the figure, she seems to press the posterior medial portion of the elbow, not the ulnar tunnel (more anterior).

  •  Good point, this picture is taken after she had returned to work ‘showing how she worked with her laptop when symptoms significantly affected her activities of daily living and work.’ To overcome this misunderstanding, we added ‘after she had returned to work’, page 4, line 133.

Publishing case report is difficult because differential should be ruled extensively, as well as getting compensation for patient.

  •  We agree. Therefore, in the newly formulated first paragraph in the discussion, we have addressed that this case description is based on the Dutch criteria for occupational disease reporting, including case definition, exposure criteria and assessment procedure and that the Dutch approach might differ from other countries, page 5, line 172-180.

Minor typo=> L91[ ….. -25] not [ …..-25)

  •  Thank you and we amended the text accordingly, page 3, line 92.

Reviewer 2 Report

The paper is generally well written in a case format with a concise and precise written style. It is easy to follow and presents a useful case how in its application country the establishment of an occupational illness case goes through and works.  The case report is of value in sharing how in its country workers and regulatory bodies reacts to an occupational claim and how an occupational illness is established.  I recommend publication except a minor rephrasing and better arguments of what are presented in it step two (evidence in literature).  Since the paper is written in a brief format, please consider the readers in the workplace who are not so deeply involved with epidemiology and more explanation helps to understand the authors use the evidence in the literature to establish the case.   Please see as follows.

The main literature evidence on work-relatedness presented in step two is based on two study results.  Reference [19] is more strongly evident, however, the evidence from [16] appears not so convincing in that the current statement in step two seems to present two somewhat conflicted (at least on the surface) findings.  It first indicates that [16] showed negative association between daily hours of computer use and ulnar neuropathy or ulnar neuropathy-like symptoms.  This implies the longer you work with computers the lower the risk.  This seems negative if one is trying establish a case between computer use and neuropathy.  In step two, it then indicates “in contrast, leaning on the elbow while working showed a significant high risk for clinically assessed ulnar neuropathy (OR= 2.16)”.   If leaning on elbow while working with computers is significant, why the longer the work with computers, the better off you are?  This also makes me wonder, why these two findings seem so “in contrast”?  This second finding of [16] is the main leverage the present assessment utilizes.  It is quite important to the case establishment of the present case study.  I then read [16] and tried to dig deeper.  As in the last point of discussion [16] , there is a precaution stated, “The negative findings on hours of daily computer use and both outcomes are in line with the studies of computer use and carpal tunnel syndrome [10,11], and up to now no studies indicate that computer use could be detrimental for the nerves in the arm or hand. From this study preventive efforts in computer work should be reserved for those leaning on an elbow while working, most often the non-dominant hand while using mouse devices”.  The main literature apparently indicates inconclusive relationship between computer use and work-related illnesses, at least in case of carpal tunnel syndrome.  It requires a better justification to this overwhelmed fact of no relationship in these systematic reviews.  It seems to me that [16] gives us an important message that the computer work may not be associated with neuropathy, but if you use them in a way that confines the posture of elbow leaning on the surface, it becomes risky.  I think this case report goes in line with this particular finding in [16].  Instead of using “in contrast”, which negate the importance of that finding, it requires a more elaborate and delicate statements and arguments to describe that particular result from [16] to avoid confusions.

[10] Thomsen JF, Gerr F, Atroshi I (2008) Carpal tunnel syndrome and the use of computer mouse and keyboard: a systematic review. BMC Musculoskeletal Disorders 9 (2008),134.

[11] Van Rijn RM, Huisstede BM, Koes BW, Burdorf A (2009) Associations between work-related factors and the carpal tunnel syndrome--a systematic review. Scand J Work Environ Health 35 (2009), 19-36.

Author Response

Reviewer 2 report:
Reviewer 2

The paper is generally well written in a case format with a concise and precise written style. It is easy to follow and presents a useful case how in its application country the establishment of an occupational illness case goes through and works. The case report is of value in sharing how in its country workers and regulatory bodies reacts to an occupational claim and how an occupational illness is  established. I recommend publication except a minor rephrasing and better arguments of what are presented in it step two (evidence in literature). Since the paper is written in a brief format, please consider the readers in the workplace who are not so deeply involved with epidemiology and more explanation helps to understand the authors use the evidence in the literature to establish the case. Please see as follows.

  •  First of all, we like to thank the reviewer for the time taken to critically review our paper and the compliments. The useful comments have improved the quality of our paper.

The main literature evidence on work-relatedness presented in step two is based on two study results. Reference [19] is more strongly evident, however, the evidence from [16] appears not so convincing in that the current statement in step two seems to present two somewhat conflicted (at least on the surface)
findings. It first indicates that [16] showed negative association between daily hours of computer use and ulnar neuropathy or ulnar neuropathy-like symptoms. This implies the longer you work with computers the lower the risk. This seems negative if one is trying establish a case between computer use and neuropathy. In step two, it then indicates “in contrast, leaning on the elbow while working showed a significant high risk for clinically assessed ulnar neuropathy (OR= 2.16)”. If leaning on elbow while working with computers is significant, why the longer the work with computers, the better off you are? This also makes me wonder, why these two findings seem so “in contrast”? This second finding of [16] is the main leverage the present assessment utilizes. It is quite important to the case establishment of the present case study. I then read [16] and tried to dig deeper. As in the last point of discussion [16] , there is a precaution stated, “The negative findings on hours of daily computer use and both outcomes are in line with the studies of computer use and carpal tunnel syndrome [10,11], and up to now no studies indicate that computer use could be detrimental for the nerves in the arm or hand. From this study preventive efforts in computer work should be reserved for those leaning on an elbow while working, most often the non-dominant hand while using mouse devices”. The main literature apparently indicates inconclusive relationship between computer use and work-related illnesses, at least in case of carpal tunnel syndrome. It requires a better justification to this overwhelmed fact of no relationship in these systematic reviews. It seems to me that [16] gives us an important message that the computer work may not be associated with neuropathy, but if you use them in a
way that confines the posture of elbow leaning on the surface, it becomes risky. I think this case report goes in line with this particular finding in [16]. Instead of using “in contrast”, which negate the importance of that finding, it requires a more elaborate and delicate statements and arguments to describe that particular result
from [16] to avoid confusions.

[10] Thomsen JF, Gerr F, Atroshi I (2008) Carpal tunnel syndrome and the use of computer mouse and keyboard: a systematic review. BMC Musculoskeletal Disorders 9 (2008),134.

[11] Van Rijn RM, Huisstede BM, Koes BW, Burdorf A (2009) Associations between work-related factors and the carpal tunnel syndrome--a systematic review. Scand J Work Environ Health 35 (2009), 19-36.

  •  Point well taken. Indeed leaning on the elbows, while working with the laptop instead of working with a standard computer, appears to be the main risk factor in our case description. Unfortunately, reference [16] does not provide a rationale for this seemingly contrary finding between more hours of computer work and a smaller risk of ulnar neuropathy. A best guess might be that workers with prolonged hours of computer work have better ‘computer ergonomics’. The importance of leaning on the elbows is also
    supported by the third study we found while updating our review to December 3rd 2020. We have amended step 2, page 3 line 108-119 and also the conclusion, page 6 line 199 to overcome this misunderstanding. For the sake of clarity, the findings regarding carpal tunnel syndrome at the wrist might of course deviate from those of ulnar neuropathy at the elbow. Also in line with the comments of the third reviewer, we added data about other neurological and musculoskeletal disorders associated with laptop use in the discussion section, page 6, line 186-190. Remarkably that recent review [33] shows that little is known about laptop use and these disorders.

Reviewer 3 Report

This is an interesting case study and the paper is well organised. There are some issues to be addressed: 

  1. The english can be improved. Comments are made in the attached pdf file of the manuscript. 
  2. In the results section, for Step 1, Diagnosis, it should be more clearly explained all the investigations conducted on this patient in more details. In the clinical examination, if there is loss of muscle strength, there should be more detailed muscle testing of all the fingers, especially IV and V, these information can be presented in a table form.  Sensation loss was described as "subjectively reported". Proper neurological examination results can be presented in a table and a figure showing the sensation loss? 
  3. "The clinical diagnosis was confirmed by electroneurography (ENG), which showed a severe conduction block across both elbows." This is very vague, was the ENG tested on the ulnar nerve? The exact results of the ENG testing can be elaborated in a more technical manner. e.g. the testing results showed ??% block of nerve conduction. What type of nerve injury was indicated in the ENG report? e.g neurotmesis, neuropraxia? 
  4. In addition, any other investigations conducted such as x rays? MRI? 
  5. Step 3 - for her previous 4 years, was a desktop computer used? did her work time in using the computer changed since switching to flexible work space? If she was moving between different work spaces, why did she always have her elbows pressing on the desk? did she ever try working in standing? 
  6. With using the laptop computer, did she also have neck and shoulder pain? if her typing posture involves pressing the medial aspects of the elbows on the desk, the shoulders would have to be abducted and the neck in a flexed posture. Anyway, ergonomic modifications and patient education are definitely needed in this case. 
  7. What are her other job duties beside working on the computer? 
  8. In Discussion, I think it should be pointed out the common upper limb MSD problems associated with prolonged computer use include carpal tunnel syndrome, neck-shoulder pain, tenosynoviits, lateral epicondylitis etc. These upper limb problems are more likely to occur compared to those lower limb conditions. 

Author Response

Reviewer 3 report
Comments and Suggestions for Authors
This is an interesting case study and the paper is well organised.

  •  First of all, we like to thank the reviewer for the time taken to critically review our paper and the compliment given. The useful comments have improved the quality of our paper.

There are some issues to be addressed:

1. The english can be improved. Comments are made in the attached pdf file of the manuscript.

  •  Thank you for your detailed suggestions and we have amended the paper accordingly, page 1 lines 14, 15 and 18.

1. In the results section, for Step 1, Diagnosis, it should be more clearly explained all the investigations conducted on this patient in more details. In the clinical examination, if there is loss of muscle strength, there should be more detailed muscle testing of all the fingers, especially IV and V, these information can be presented in a table form. Sensation loss was described as "subjectively reported". Proper neurological examination results can be presented in a table and a figure showing the sensation loss?
2. "The clinical diagnosis was confirmed by electroneurography (ENG), which showed a severe conduction block across both elbows." This is very vague, was the ENG tested on the ulnar nerve? The exact results of the ENG testing can be elaborated in a more technical manner. e.g. the testing results showed ??% block of nerve conduction. What type of nerve injury was indicated in the ENG report? e.g neurotmesis, neuropraxia?
3. In addition, any other investigations conducted such as x rays? MRI?

  •  Unfortunately we are not able to provide the requested data. This is partly due to how occupational diseases are reported in the Netherland as is described in the revised discussion (page 5, line 172-180) and that the patient now receives her occupational health care form another occupational health service and given the Dutch privacy regulations we feel that it is not appropriate to contact the patient any more than already is done for this case report.

4. Step 3 - for her previous 4 years, was a desktop computer used? did her work time in using the computer changed since switching to flexible work space? If she was moving between different work spaces, why did she always have her elbows pressing on the desk? did she ever try working in
standing?

  •  Yes, a desktop computer was used. Her work time at the computer did not change. We amended the text accordingly, page 3 line 121-125. The other questions cannot be addressed given that the patient now receives her occupational health care from another occupational health service and given the Dutch privacy regulations.

5. With using the laptop computer, did she also have neck and shoulder pain? if her typing posture involves pressing the medial aspects of the elbows on the desk, the shoulders would have to be abducted and the neck in a flexed posture. Anyway, ergonomic modifications and patient education are definitely needed in this case.

  •  No other upper-limb work-related musculoskeletal disorder were present as mentioned at page 5, line 137. We totally agree with the reviewer regarding the recommendations made as mentioned at page page 6, line 196-197.

6. What are her other job duties beside working on the computer?

  •  Her job mainly consisted of computer work, page 3 line 121-125.

7. In Discussion, I think it should be pointed out the common upper limb MSD problems associated with prolonged computer use include carpal tunnel syndrome, neck-shoulder pain, tenosynoviits, lateral epicondylitis etc. These upper limb problems are more likely to occur compared to those lower limb conditions.

  •  Thank you for this good suggestion and we added lines 186-190, page 6 showing that remarkably little is known about laptop use and other neurological or musculoskeletal disorders based on a recent review [33].

Round 2

Reviewer 1 Report

I still feel it is a case report, not a review, and still missing (as CARE mentioned) better detail on other factors (including complex with imaging but also simple, like BMI because the picture seems to show an overweight person). I agree that the conclusion should be “laptop was suggested here though association is not found on literature” (and the BMI was ...). 

Author Response

I still feel it is a case report, not a review, and still missing (as CARE mentioned) better detail on other factors (including complex with imaging but also simple, like BMI because the picture seems to show an overweight person).

=> First of all, we like to thank the reviewer again for taken the time to review our revised manuscript: much appreciated. We altered the manuscript using track changes (and the highlighted yellow lines are our former revisions).

=> We agree that our paper is a case report and we have omitted the review phrase in the abstract to overcome this misunderstanding, page 1, line 14-15.

=> The diagnosis was made by a neurologist and confirmed by electroneurography, page 2, line 78-79. We hope this is sufficient.

=> As mentioned on page 5 line 135, we could rule out obesity. As already said in our previous response, the patient now receives her occupational health care from another occupational health service and given the Dutch privacy regulations we feel that it is not appropriate to contact the patient again regarding her BMI. In addition, given that the recent 2020 study [26] shows that BMI has an OR=1.05 we more cautiously formulated the presence of other risk factors: page 5, line 148-150 …  no other non-work-related or personal risk factors with a similar risk were present.

I agree that the conclusion should be “laptop was suggested here though association is not found on literature” (and the BMI was ...). 

=> Thank you and in line with your suggestion and our above mentioned remark about BMI, we added, page 6, line 199 … although not previously described in the medical literature. We also amended the conclusion in the abstract, page 1, line 17-19.

Round 3

Reviewer 1 Report

I feel it is too bad that you could not give more detail on associated factors.

If you cite the great citation of our Italian colleague, I feel you should mentioned it completely: "At multivariable analysis, UNE was associated to male gender (OR = 2.4, 95%CI = 1.6‐3.7), smoking habits (>25 pack‐years (OR = 2.3, 95%CI = 1.3‐4.1), body mass index (OR = 1.05, 95%CI 1.01‐1.10), polyneuropathies (OR = 4.1, 95%CI 1.5‐11.5), and leaning with flexed elbow on a table/desk (OR = 1.5, 95%CI 1.0‐2.2)." Polyneuropathy should have been excluded and giving the results of imaging/NCS. Second the BMI seemed low but it is a continuous variable and significant.

I might be too demanding but without this, it could be acceptable in many countries as work related.

However, I think you made the best you could with your data/ regulation and the readers might be interested and then should be accepted.